# Enhancing the Transferability of Targeted Attacks with Adversarial Perturbation Transform

**Zhengjie Deng [1,2,\*]** **, Wen Xiao [1], Xiyan Li [1], Shuqian He [1] and Yizhen Wang [3]**

[1] School of Information Science and Technology, Hainan Normal University, Haikou 571158, China; xiaowenqp@163.com (W.X.); xiyanli2006@163.com (X.L.); heshuqian05@126.com (S.H.)
[2] Guangxi Key Laboratory of Image and Graphic Intelligent Processing, Guilin 541004, China
[3] School of Physics and Electronic Engineering, Hainan Normal University, Haikou 571158, China; wangxuesu1980@163.com
[\*] Correspondence: hsdengzj@163.com

**Abstract:** The transferability of adversarial examples has been proven to be a potent tool for successful attacks on target models, even in challenging black-box environments. However, the majority of current research focuses on non-targeted attacks, making it arduous to enhance the transferability of targeted attacks using traditional methods. This paper identifies a crucial issue in existing gradient iteration algorithms that generate adversarial perturbations in a fixed manner. These perturbations have a detrimental impact on subsequent gradient computations, resulting in instability of the update direction after momentum accumulation. Consequently, the transferability of adversarial examples is negatively affected. To overcome this issue, we propose an approach called Adversarial Perturbation Transform (APT) that introduces a transformation to the perturbations at each iteration. APT randomly samples clean patches from the original image and replaces the corresponding patches in the iterative output image. This transformed image is then used to compute the next momentum. In addition, APT could seamlessly integrate with other iterative gradient-based algorithms, incurring minimal additional computational overhead. Experimental results demonstrate that APT significantly enhances the transferability of targeted attacks when combined with traditional methods. Our approach achieves this improvement while maintaining computational efficiency.

**Keywords:** adversarial examples; transferability; perturbation transform; targeted attacks

## 1. Introduction

In recent years, deep learning has rapidly developed and found applications in various fields, such as autonomous driving [1,2] and face recognition [3,4]. However, deep neural networks (DNNs) face the threat of adversarial examples [5,6]. Attackers can fool DNNs by adding some imperceptible disturbances to the input images. Given the susceptibility of neural networks to adversarial examples, DNNs encounter significant challenges in real-world applications. Several defenses [7–11] have been proposed to defend against adversarial examples. Additionally, numerous adversarial attack methods [6,7,12,13] have been developed to evaluate the robustness of DNNs.

However, in the real world, it is difficult for attackers to access all the information of the target model. So, white-box attack methods [6,7,12,13] are hard to apply in realistic scenarios. With decision-based black-box attacks [14,15], attackers do not need to grasp the internal structure of the model and have a certain viability, but it will take a large number of queries and time consumption. This will trigger the security system's vigilance. If the defender limits the number of queries, the black-box attack will not succeed. The transferability of adversarial examples enables real-world attacks in black-box scenarios, where attackers lack knowledge of the model's structure and parameters and are not required to query the target model.

In general, attack methods can be divided into two categories: targeted and non-targeted attacks. Targeted attacks aim to misidentify adversarial examples as a specific class, while non-targeted attacks focus on decreasing the accuracy of the victim model. Recent studies have proposed methods to enhance the transferability of non-targeted attacks [16–20], with the objective of reducing the accuracy of the target model. However, the success rate of transferable targeted attacks, where the attackers must deceive the victim model to produce a predetermined specific outcome, still remains lower than that of non-targeted attacks.

Overfitting to the source model is a primary factor contributing to the limited transferability of adversarial examples. To address this issue, several techniques have been proposed to enhance the transferability of adversarial examples. These techniques include input transformations [21], translation invariant attacks [22], advanced gradient computation [17,22,23], and advanced loss functions [24,25]. Among these, input transformation stands out as one of the most effective methods, drawing inspiration from data augmentation techniques used in model training [26]. By applying diverse transformations to input images, this approach aims to prevent adversarial examples from overfitting to the proxy model. However, most existing methods focus solely on diversifying individual input images, and these fixed transformations may still overfit to the internal environment of the source model, rendering them unsuitable for unknown black-box models.

Existing gradient iteration algorithms, such as MI-FGSM [17], have limitations in terms of generating adversarial perturbations with high transferability. These limitations arise due to the excessive consistency and redundancy in the perturbations generated during the iterative attack process, which is constrained by a perturbation budget. This consistency leads to a high degree of redundancy between successive perturbations, resulting in momentum accumulation in a fixed direction. As a result, the update direction becomes unstable, negatively impacting the transferability of adversarial examples.

The proposed Adversarial Perturbation Transform (APT) method aims to overcome these limitations and enhance the transferability of targeted attacks. APT randomly selects perturbed patches from the input image at each iteration and restores them to their original state, creating an admixed image. Then, this admixed image is included in the gradient calculation process. This approach effectively prevents excessive consistency and redundancy between successive perturbations, leading to a more stable update direction.

In summary, our main contributions are as follows:

- We identify the redundancy and consistency of adversarial perturbations generated during iterative attacks. They have a negative impact on the stability of the update direction.
- We introduce the inclusion of clean patches from the original input image to bootstrap the computation of gradients, presenting an effective method for enhancing the transferability of targeted attacks.
- Our method is compatible with most algorithms based on MI-FGSM and incurs minimal additional computational overhead. Empirical evaluations demonstrate that APT significantly improves the transferability of targeted attacks.

## 2. Related Work

The vulnerability of deep neural networks (DNNs) to adversarial examples was first mentioned by [5]. These adversarial images can exploit the vulnerability of the models to fool them, inducing the model to classify the samples into the wrong classes with high probability.

Szegedy et al. [5] initially used the LBFGS to generate adversarial examples. Due to the high computational cost, Goodfellow et al. [6] proposed the fast gradient sign method (FGSM), which effectively generates adversarial examples by performing a single gradient step. Kurakin et al. [27] used an iterative method ( I-FGSM ) to extend FGSM. To avoid local minima to improve transferability, ref. [17] incorporated the momentum iterative gradient method to boost the transferability effect of the generated adversarial examples.

Several techniques have been proposed to improve transferability by helping the image avoid falling into local minima and prevent overfitting to the specific source model. DI (Diverse Input) [21] randomly resizes and fills the image for each input. Zou et al. [28] introduced a three-stage pipeline, resizing diverse input, diversity ensemble, and region fitting, which work together to enhance the transferability. TI (Translation Invariant) [29] generates several translated versions of the current image and uses a convolution to approximate the gradient fusion. SI (Scale Invariant) [22] exploits the scale-invariant property of CNNs and uses multiple scale copies from each input image. Admix [30] randomly samples a set of images from other classes and computes the gradient of the original image mixed with a small portion of additional images while using the original labels of the input to make more transferable examples. Reference [31] trains a CNN as an adversarial transformation network, which neutralizes the adversarial perturbations and thus constructs more powerful adversarial examples. ODI (object-based diverse input) [32] renders images onto different 3D target objects and classifies the rendered objects into target classes, including different lighting and viewpoints, while it also requires additional computational overhead.

Lin et al. [22] incorporated the Nesterov accelerated gradient method into an iterative gradient-based attack to mitigate the issue of local optima in the optimization process. Wang et al. [23] addressed the instability of the update direction by considering the gradient variance from the previous iteration. They adjusted the current gradient based on this variance, which helped stabilize the update direction and improved the performance of the attack. The choice of loss function also plays a significant role in the effectiveness of targeted attacks. Li et al. [24] observed that using cross-entropy loss (CE) can lead to gradient vanishing during an attack. To increase targeted transferability, they proposed using Poincare distance as the loss function. However, Zhao et al. [25] argued that using Poincare distance can result in a large step size and a coarse loss function surface, leading to worse results compared to cross-entropy loss in different architecture models. They suggested using a simple logit loss for targeted attacks and emphasized the importance of conducting a sufficient number of iterations to generate effective adversarial examples.

## 3. Methodology

This paper focuses on the transferability of targeted attacks. It firstly describes the attack target and notation, then introduces details of the APT method and the motivation.

### 3.1. Preliminary

The attack method for generating adversarial examples can be considered an optimization problem. Let $x$ be the input image with the ground-truth label $y$. $f(\cdot)$ indicates the neural network classifier, $J$ be the loss function, and generated adversarial examples $x^{\mathrm{adv}}$ with target label $y^{\mathrm{target}}$. And the perturbation is constrained by the $\ell_\infty$-norm which can be formulated as $\left\| x^{\mathrm{adv}} - x \right\|_\infty \leq \epsilon$. Then, the targeted attack aims to solve the following optimization problem, presented by Equation (1).

$$\arg\min_{x^{\mathrm{adv}}} J\left(f(x^{\mathrm{adv}}), y^{\mathrm{target}}\right)$$
$$\text{s.t. } \left\| x^{\mathrm{adv}} - x \right\|_\infty \leq \epsilon. \tag{1}$$

Here, we use the logit loss as the loss function [25]. The formulation is Equation (2).

$$J\left(f(x^{\mathrm{adv}}), y^{\mathrm{target}}\right) = -l_{\mathrm{target}}(f(x)), \tag{2}$$

where $l_{\mathrm{target}}(.)$ denotes the logit output with respect to the target class.

### 3.2. Adversarial Perturbation Transform

Adversarial Perturbation Transform (APT) is a data enhancement technique that involves the following steps: Firstly, the input image $x$ is divided into several patches denoted as $\Omega = \{u_0, u_1, \ldots, u_i, \ldots, u_m\}_{i=0}^m$, where $m$ represents the total amount of patches in the image and $u_i \in \mathbb{R}^{3 \times n \times n}$ with $n$ being the dimensions of each patch. Secondly, during each iteration, a proportion $p$ of clean patches is randomly sampled from the input image to obtain $\xi_{p,t} \subset \Omega$. This process results in an image $x_t^{\text{clean}}$ with $p \cdot m$ clean patches, where $t$ denotes the t-th iteration. Mathematically, this can be represented as

$$x_t^{\text{clean}} = x \odot R_t^p \tag{3}$$

In Equation (3), the Hadamard product is denoted by the symbol $\odot$. We use a binary matrix $R_t$ in our algorithm, which consists of tiny matrices with a randomly distributed pattern of 0 and 1. The proportion of 1 s in $R_t$ is represented by $p$. In the next step, we remove the patches from the perturbed images that correspond to the 0 elements in $R_t$. This results in an image $x_t^{\text{perturb}}$ with only $(1 - p) \cdot m$ perturbed patches. Mathematically, we achieve this using the following equation:

$$x_t^{\text{perturb}} = x_t^{\text{adv}} \odot (I - R_t^p) \tag{4}$$

In Equation (4), $I$ denotes the matrix with all elements equal to 1. In the final step, we combine $x^{\text{clean}}$ and $x^{\text{perturb}}$ into a composite image, which consists of $p \cdot m$ clean patches and $(1 - p) \cdot m$ perturbed patches. This fusion process can be achieved using the following equation, Equation (5):

$$x_t^{\text{mix}} = x_t^{\text{clean}} + x_t^{\text{perturb}} \tag{5}$$

Finally, $x_t^{\text{mix}}$ is fed into the proxy model to participate in the gradient calculation, and we show the APT method involved in the MI-FGSM [17] attack process as an example in Algorithm 1.

---

**Algorithm 1** APT-MI-FGSM

---

**Require:**　a classifier $f$; the logit loss function $J$;
**Require:**　a clean image $x$; the target label $y^{\text{target}}$;
**Require:**　iterations T; decay factor $\mu$; the max perturbation $\epsilon$; step size $\alpha$.
**Ensure:**　an adversarial example $x^{\text{adv}}$
1: $g_0 = 0; x_0^{\text{adv}} = x$
2: **for** $t = 0 \rightarrow T - 1$ **do**
3:　　generate the mix image $x_t^{\text{mix}}$ by Equations (3)–(5)
4:　　calculate the gradient $\hat{g} = \nabla_{x_t^{\text{adv}}} J\big(f(x_t^{\text{mix}}), y^{\text{target}}\big)$
5:　　update $g_{t+1} = \mu \cdot g_t + \hat{g} / \|\hat{g}\|_1$
6:　　update $x_{t+1}^{\text{adv}} = x_t^{\text{adv}} - \alpha \cdot sign(g_{t+1})$
7: **end for**
8: $x^{\text{adv}} = x_T^{\text{adv}}$
9: **return** $x^{\text{adv}}$

---

### 3.3. Motivations

Most existing gradient optimization methods [21,28,29,32] utilize MI-FGSM [17] as the baseline, which introduces momentum in the iterative attack process. This momentum helps to stabilize the update direction and prevents the convergence to poorer local maxima. However, generating more robust target adversarial examples requires a sufficient number of iterations [25]. The fixed perturbation input leads to redundancy during momentum accumulation, limiting the effectiveness within the perturbation budget. Figure 1 presents the gradient and momentum cosine similarity between iterative attacks. Figure 1a illustrates

cases where the cosine similarity between two successive gradients is less than 0 when using MI-FGSM, indicating opposite directions. Consequently, the momentum after gradient accumulation does not converge sufficiently, resulting in an unstable update direction (Figure 1b).

Data augmentation is an effective technique for enhancing the transferability of adversarial examples [21,22,28–30,32]. However, most existing input transformation methods focus on a single perturbed input image with relatively fixed transformations. While these methods alleviate over-fitting to some extent, the issue of gradient redundancy persists. DI-TI [21,29] transformation partially addresses the gradient direction problem (Figure 1c). However, part of the gradients in consecutive calculations remain highly similar, and the accumulated momentum is still insufficient for convergence (Figure 1d). The algorithm improved by APT makes the two continuous gradients maintain a low similarity, which ensures the momentum's direction is more stable.

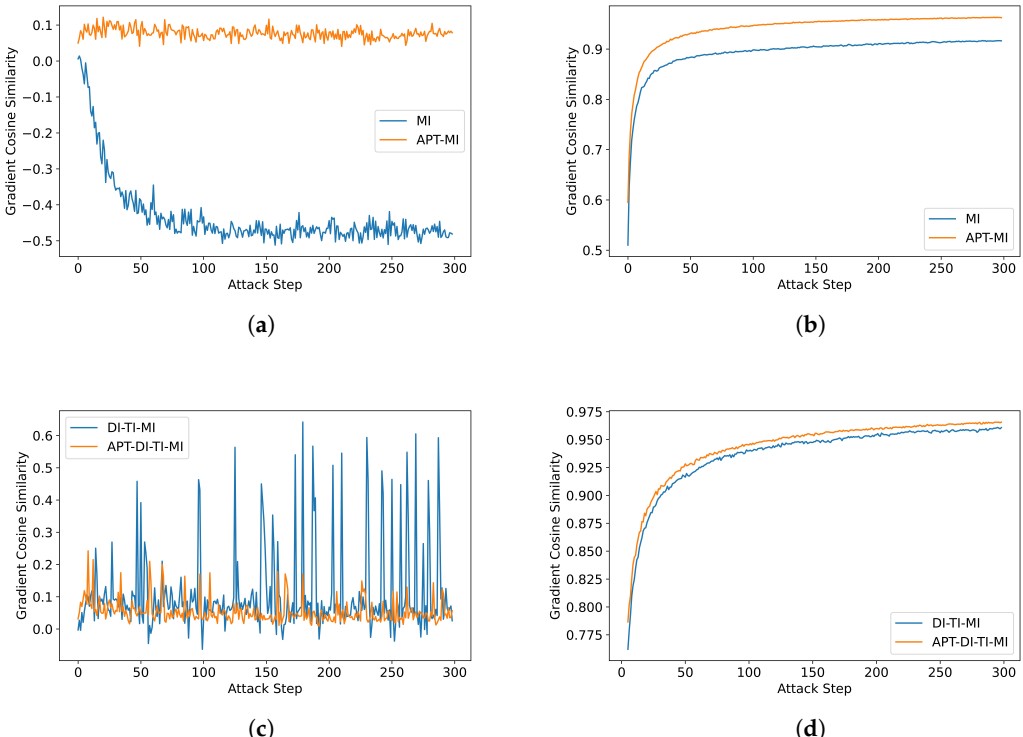

**Figure 1.** Analysis of gradient and momentum cosine similarity between iterative attacks: (**a**) Cosine similarity between two successive gradients. (**b**) Cosine similarity between two successive momentums. (**c**) Cosine similarity between two successive gradients. (**d**) Cosine similarity between two successive momentums.

The transferability of adversarial examples can be seen as analogous to generalization in model training. While the latter trains a robust model for classification, the former aims to train a robust sample capable of successfully attacking various models. To simulate different unknown black-box environments, we introduce perturbation distortion by eliminating some perturbations and require the generated adversarial examples to resist this distortion.

We address the issue of gradient redundancy by randomly selecting reduced adversarial patches, making adjacent iterations of the input perturbation completely different. Calculating gradients for mixed patches has the potential to enhance the transferability of targeted attacks. It is important to note that APT differs from SI and Admix methods in that APT only changes part of the image, while SI and Admix change the whole image and generate multiple copies to calculate so that APT is more efficient.

## 4. Experiment

### 4.1. Experimental Settings

**Dataset.** We used a dataset compatible with ImageNet from the NIPS2017 Adversarial Attack and Defense Competition [33]. The dataset consists of 1000 images with dimensions $299 \times 299$ and labels corresponding to ImageNet's target class tasks for targeted attacks.

**Models.** We evaluated both undefended models (trained normally) and defended models as target models. For the undefended models, we selected Resnet50 (Res50) [34], DenseNet121 (Den121) [35], VGG16 [36], Inception-v3 (Inc-v3) [37], MobileNet-v2 (Mob-v2) [38], and Inception-Resnet-v2 (IR-v2) with different architectures. As for the defended models, we chose three adversarially trained models [39]: ens3-adv-Inception-v3 (Inc-v3-ens3), ens4-adv-Inception-v3 (Inc-v3-ens4), and ens-adv-Inception-Resnet-v2 (IR-v2-ens4).

**Baselines.** Since a simple transformation alone does not yield satisfactory results in terms of targeted transferability, we employed a composite method that combines several classical algorithms, namely MI [17], DI [21], RDI [28], ODI [32], and TI [29], as baselines for comparison.

**Implementation details.** We set the transformation probability $p_{DI}$ to 0.7 in DI and randomly enlarged the image size within the range of $[299, 330]$. The kernel convolution size in TI was set to 5. Regarding the hyperparameters used in the iterative process, we set the decay factor $\mu$ to 1, the step size $\alpha$ to $2/255$, the number of iterations to 300, and the perturbation size to $\epsilon = 16/255$. In the experiment evaluating APT attacks, we set the patch size $n$ to 10 and the hyperparameter $p$ to 0.3.

**Evaluation Metrics.** We measured the success rate of targeted attacks (suc), which indicates the percentage of instances where the black-box model was fooled into predicting the specified category.

### 4.2. Attacking Naturally Trained Models

In reality, attackers have no access to grasp all the information of the target model. Therefore, we set up experiments between different architectural models. This assumption aligns more with the real-world scenario.

The results of the APT with baseline attacks are shown in Table 1. From the results, APT is very effective in enhancing the transferability in targeted attacks. Taking ResNet50 as source model for example, the average performance improvements induced by APT are 54.0% (DI-TI), 20.78% (RDI-TI), and 3.7% (ODI-TI), respectively. Comparing to the ResNet50 and DenseNet121, all attacks generally achieve lower transferability when using VGG-16 or Inception-v3 as the source models. This may be explained by the fact that skip connections in ResNet50 and DenseNet121 improve the transferability [40]. APT's improvement of the effectiveness of adversarial attacks using weak transferability models is more obvious. For example, when Inception-3 was the source model, APT improved the average performance with 246.2% (DI-TI), 125.8%(RDI-TI), and 32.3% (ODI-TI). However, for DenseNet121 and VGG16 as the source models, APT also consistently boosted the transferability under all cases.

In Figure 2, we present visual comparisons of the adversarial examples generated by the DI-TI and APT-DI-TI attacks, using ResNet50 as the proxy model. The results demonstrate that there is little difference in the degree of perturbations between the adversarial examples generated by the two methods.

**Table 1.** Targeted fooling rates (%) of different attacks against various architectural models. The best results are highlighted in blue. * Indicates white-box attacks.

| Source | Attack | Res50 | Den121 | VGG16 | Inc-v3 | Mob-v2 | IR-v2 | Average |
|--------|--------|-------|--------|-------|--------|--------|-------|---------|
| Res50 | DI-TI | 98.7 * | 72.8 | 61.8 | 8.7 | 28.5 | 13.7 | 37.1 |
| | RDI-TI | 97.6 * | 82.0 | 67.3 | 30.3 | 45.3 | 39.8 | 52.94 |
| | ODI-TI | 99.1 * | 85.1 | 75.6 | 56.7 | 57.4 | 60.0 | 66.96 |
| | APT-DI-TI | 98.7 * | 85.0 | 80.0 | 28.2 | 55.9 | 36.7 | 57.16 |
| | APT-RDI-TI | 97.9 * | 82.1 | 73.2 | 46.6 | 62.0 | 55.8 | 63.94 |
| | APT-ODI-TI | 98.6 * | 84.2 | 76.6 | 58.1 | 66.1 | 61.7 | 69.4 |
| Den121 | DI-TI | 44.7 | 98.9 * | 38.0 | 7.8 | 13.7 | 10.4 | 22.92 |
| | RDI-TI | 56.5 | 98.6 * | 41.7 | 22.0 | 20.9 | 29.9 | 34.2 |
| | ODI-TI | 68.4 | 98.8 * | 62.5 | 44.9 | 36.2 | 51.7 | 52.74 |
| | APT-DI-TI | 68.8 | 98.1 * | 63.4 | 22.3 | 31.7 | 29.9 | 43.22 |
| | APT-RDI-TI | 67.7 | 97.3 * | 57.9 | 36.6 | 37.9 | 47.1 | 48.36 |
| | APT-ODI-TI | 70.5 | 98.2 * | 65.6 | 48.1 | 48.3 | 53.1 | 57.12 |
| VGG16 | DI-TI | 10.2 | 14.7 | 95.0 * | 0.7 | 5.1 | 0.5 | 6.24 |
| | RDI-TI | 30.2 | 34.8 | 94.7 * | 6.6 | 19.5 | 7.9 | 19.8 |
| | ODI-TI | 50.7 | 61.7 | 95.2 * | 22.3 | 34.3 | 26.9 | 39.18 |
| | APT-DI-TI | 26.0 | 29.2 | 95.0 * | 3.4 | 14.1 | 4.3 | 15.4 |
| | APT-RDI-TI | 41.0 | 45.3 | 95.4 * | 11.4 | 28.4 | 12.8 | 27.8 |
| | APT-ODI-TI | 55.4 | 63.5 | 94.8 * | 29.3 | 43.3 | 30.5 | 44.4 |
| Inc-v3 | DI-TI | 2.5 | 5.0 | 2.8 | 99.0 * | 1.8 | 9.1 | 4.24 |
| | RDI-TI | 4.7 | 6.7 | 4.1 | 99.1 * | 2.3 | 14.0 | 6.36 |
| | ODI-TI | 14.5 | 23.7 | 11.3 | 99.1 * | 9.2 | 37.8 | 19.3 |
| | APT-DI-TI | 10.1 | 17.1 | 10.1 | 98.7 * | 7.5 | 28.6 | 14.68 |
| | APT-RDI-TI | 9.9 | 14.4 | 8.4 | 98.2 * | 8.3 | 30.8 | 14.36 |
| | APT-ODI-TI | 19.5 | 30.0 | 18.3 | 98.5 * | 15.3 | 44.6 | 25.54 |

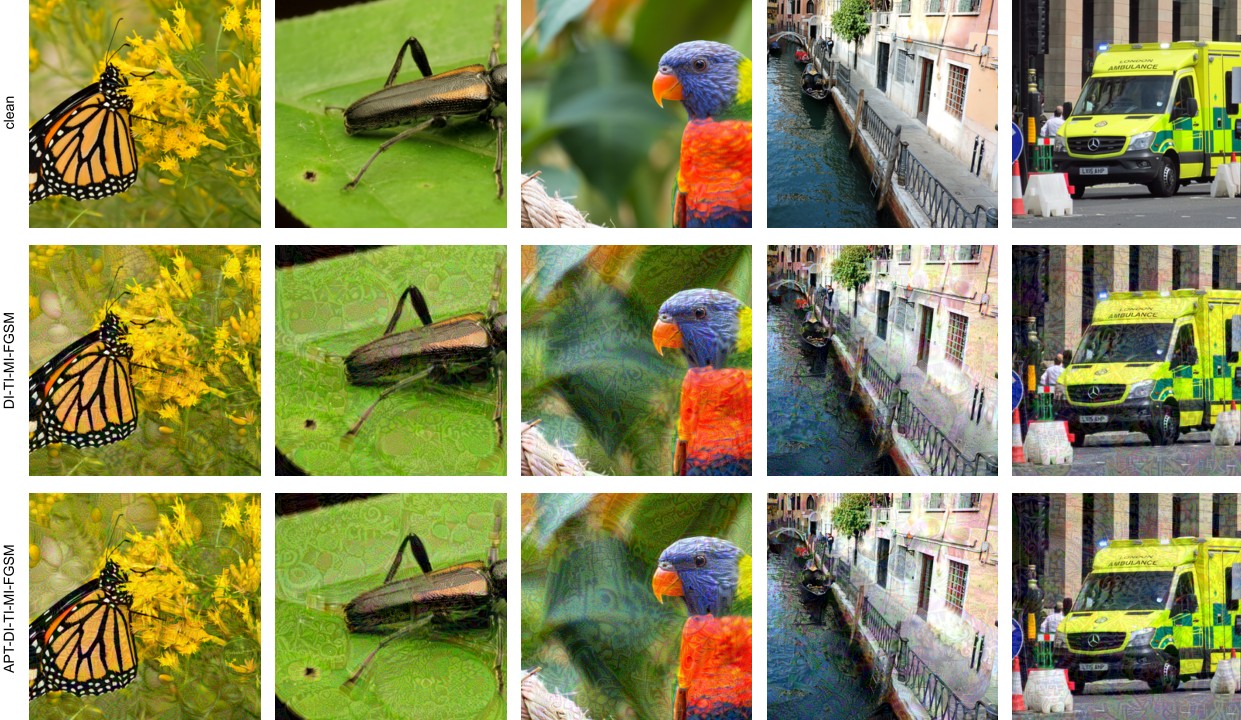

**Figure 2.** Visualization of adversarial images generated on ResNet50.

### 4.3. Attacking Adversarially Trained Models

Adversarial training, proposed by Tramer et al. [39], is widely recognized as an effective defense method against adversarial attacks, especially targeted attacks. Fooling adversarially trained models into predicting the target class successfully is a challenging task. In this study, we employed an ensemble of white-box models, including Resnet50, DenseNet121, VGG16, and Inception-v3, which were trained naturally. The results presented in Table 2 illustrate the success rates of attacks on the adversarially trained models. Our findings demonstrate that the adversarially trained models effectively resisted the adversarial examples. However, APT increased the average success rate of the state-of-the-art (SOTA) method from 1.13% to 5.56%, indicating that adversarially trained defense models may still be vulnerable to adversarial attacks.

**Table 2.** Targeted fooling rates (%) of black-box attacks against three defense models under multi-model setting. Best results are highlighted in blue.

| Attack | Inc-v3-ens3 | Inc-v3-ens4 | IR-v2-ens4 |
|:---:|:---:|:---:|:---:|
| DI-TI | 0.0 | 0.0 | 0.0 |
| RDI-TI | 0.3 | 0.1 | 0.1 |
| ODI-TI | 1.5 | 1.6 | 0.3 |
| APT-DI-TI | 0.1 | 0.1 | 0.0 |
| APT-RDI-TI | 2.0 | 2.1 | 0.4 |
| APT-ODI-TI | 8.2 | 6.6 | 1.9 |

### 4.4. Ablation Study

In this section, we conduct ablation experiments to study the influence of the hyperparameter $p$ on the performance of three algorithms and different proxy models. These experiments enable an investigation into the sensitivity of our proposed method to the hyperparameter $p$ and offer insights into the optimal value of $p$ for achieving the highest performance.

Figure 3 demonstrates the average success rates of the three distinct methods across various hyperparameter values of $p$. Notably, APT-DI-TI, APT-RDI-TI, and APT-ODI-TI correspond to DI-TI, RDI-TI, and ODI-TI when $p = 0$, respectively. Upon observation, it becomes apparent that discarding a small portion of adversarial patches leads to a noteworthy improvement in transferability. Nevertheless, an excessive discarding of patches results in a substantial loss of gradient information, consequently leading to a reduction in transferability. The findings indicate that the optimal value of $p$ depends on the specific baseline algorithm employed. Specifically, APT-DI-TI achieved the highest performance at $p = 0.4$, APT-RDI-TI exhibited the best performance at $p = 0.3$, and APT-ODI-TI performed optimally at $p = 0.2$. The results show that the value of $p$ is very critical when using different baseline algorithms.

In addition, we observed variations in the selection of the optimal $p$ value for different proxy models. Considering APT-DI-TI as the study object, it achieved the highest performance with $p = 0.3$ when Resnet 50 was the source model. For Densenet21 or VGG16 as the proxy model, the optimal $p$ value increased to $p = 0.4$. Furthermore, if Inception-v3 was the proxy model, APT-DI-TI performed best with $p = 0.5$. These results highlight the close relationship between proxy model selection and adjusting $p$, offering valuable guidance for further performance optimization.

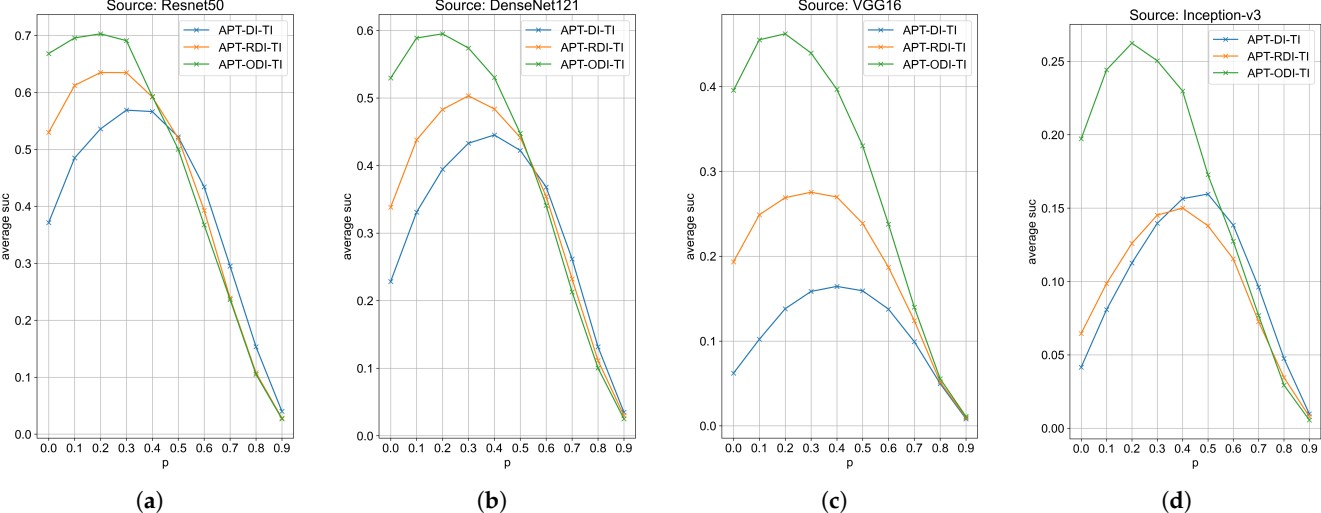

**Figure 3.** Impact of *p* on targeted transferability between different architectural models. (**a**) Targeted attack success rates under different hyperparameters *p* against the source model Resnet50. (**b**) Targeted attack success rates under different hyperparameters *p* against the source model DenseNet121. (**c**) Targeted attack success rates under different hyperparameters *p* against the source model VGG16. (**d**) Targeted attack success rates under different hyperparameters *p* against the source model Inception-v3.

## 5. Conclusions

In this paper, we identify the limitations of current gradient iteration algorithms for targeted attacks. We introduce a novel approach by considering the gradient information of clean patches in the image and proposing an improved method to enhance the transferability of adversarial examples. Our extensive experiments on ImageNet demonstrate that our proposed method, APT, achieves significantly higher success rates against black-box models compared to traditional attack methods. As a result, our method can serve as an effective benchmark for evaluating future defense mechanisms. For future work, we intend to consider other transforms and to investigate theoretical explanations for the high transferability of targeted attacks of perturbations.

**Author Contributions:** Formal analysis, project administration, writing—review and editing, and supervision, Z.D.; methodology, validation, formal analysis, writing—original draft preparation, and investigation, W.X.; validation and writing—review and editing, X.L.; data curation and supervision, S.H.; writing—review and editing, Y.W. All authors have read and agreed to the published version of the manuscript.

**Funding:** This work was financially supported by Natural Science Foundation of Hainan Province (No. 620RC604, No. 623QN236), Hainan Province Higher Education Teaching Reform Research Funding Project(No. Hnjg2023-49, Hnjg2021-37, Hnjg2020ZD-14), the Science and Technology Project of Haikou (No. 2022-007), the Open Funds from Guilin University of Electronic Technology, Guangxi Key Laboratory of Image and Graphic Intelligent Processing (No. GIIP2012), 2022 Hainan Normal University's Graduate Innovation Research Project(No. hsyx2022-91), the Hainan Province Key R&D Program Project (No. ZDYF2021GXJS010), the Major Science and Technology Project of Haikou City (No.2020006).

**Data Availability Statement:** Image data can be obtained at https://github.com/cleverhans-lab/cleverhans/ tree/master/cleverhans_v3.1.0/examples/nips17_ adversarial_competition/dataset (accessed on 10 June 2023).

**Conflicts of Interest:** The authors declare no conflict of interest.

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
