# Peer review of "Enhancing the Transferability of Targeted Attacks with Adversarial Perturbation Transform"

_electronics, doi:10.3390/electronics12183895_

Round 1

Reviewer 1 Report

Please see the attached file report

Minor editing 

Reviewer 2 Report

In this paper, the authors propose the Adversarial perturbation transform (APT) which is a data enhancement technique.

This is a very timely research topic, but at the present time it is difficult to assess what is the novel contribution of the authors.

I have the following remarks regarding methodology:

1) Isn't equation (3-5) simply a salt-pepper noise overlay operation with linear distribution? Please comment on that.

2) What was the probability of drawing 0 and 1 in (3)?

3) Is the application of the proposed method significantly different from the application of hadamard product with white noise scalled to certain range?

4) "We identify the limitations of existing gradient iterative algorithms in computing gradients." - this sentence sounds as if the authors made the identification with the help of methods of mathematical analysis, in practice they made a discussion on adversarial attacks. Please clarify the above statement.

5) Please discuss whether against adversarial attack protects the use of median or Gaussian filter?

6) Table 2 - what does nc-v3-ens3, Inc-v3-ens4 etc. mean?

7) "We used a dataset compatible with ImageNet from the NIPS2017 Adversarial Attack and Defense Competition". - this data is referenced in paper https://arxiv.org/abs/1804.00097

Please compare your results with those published in that paper.

Round 2

Reviewer 2 Report

Authors have addressed all my remarks. In my opinion paper can be accepted as it is.